

# Assessment of the Spectral MIsaLignment Effect (SMILE) on EarthCARE's Multi-spectral imager aerosol and cloud property retrievals

Nicole Docter[1,*], Anja Hünerbein[2,*], David P. Donovan[3], Rene Preusker[1], Jürgen Fischer[1], Jan Fokke Meirink[3], Piet Stammes[3], and Michael Eisinger[4]

[1]Freie Universität Berlin (FUB), Institute for Space Science, Berlin, Germany
[2]Leibniz Institute for Tropospheric Research (TROPOS), Leipzig, Germany
[3]Royal Netherlands Meteorological Institute, De Bilt, The Netherlands
[4]European Space Agency, ESA-ECSAT, Didcot, United Kingdom
[*]These authors contributed equally to this work.

**Correspondence:** Nicole Docter (nicole.docter@fu-berlin.de) and Anja Hünerbein (anjah@tropos.de)

**Abstract.** The Multi-spectral Imager (MSI) on board the Earth Cloud, Aerosol and Radiation Explorer (EarthCARE) will provide horizontal information about aerosols and clouds. These measurements are needed to extend vertical cloud and aerosol property information, which are obtained from EarthCARE's active sensors, in order to obtain a full three dimensional view on cloud and aerosol conditions. Especially, meso-scale weather systems will be characterized. The discovery of a non-compliance

of the MSI VNS camera's visible (VIS) and shortwave-infrared (SWIR1) channels regarding a spectral central wavelength (CWVL) shift across track of up to 14 nm (VIS) and 20 nm (SWIR1), led to the need for an analysis regarding its impact on MSI Level 2A aerosol and cloud products. A significant influence of the Spectral MIsaLignment Effect (SMILE) on MSI retrievals is identified due to the spectral variation of gas absorption, surface reflectance as well as aerosol and cloud properties within the spectral ranges of these MSI bands. For example, the VIS channel is positioned in close proximity to the red edge

of green vegetation and is impacted by residual absorption of water vapour and ozone. Small central wavelength variations introduce uncertainties due to the rapid change in surface reflectance for conditions with low optical thickness. The present central wavelength shift in the VIS towards shorter wavelengths than at nadir introduces a relative error in transmission of up to 3.3% due to the increasing influence of water vapour and ozone absorption. We found relative errors in the TOA signal due to the SMILE of up to 30% for low optical thickness over a land surface in that band. Since the magnitude of the impact

strongly depends on the underlying surface and atmospheric conditions, we conclude that accounting for the SMILE in Level 2 retrievals or correcting the Level 1 signal will improve MSI aerosol and cloud product quality.

## 1 Introduction

The European and Japanese Earth Cloud, Aerosol and Radiation Explorer (EarthCARE) mission aims to evaluate and enhance the representation of aerosols, clouds and precipitation in numerical weather prediction and climate models (Wehr et al.,



2023). Three of the four instruments are used to measure profile and columnar information of aerosols and clouds. These are the ATmospheric LIDar (ATLID), the Cloud profiling Radar (CPR) and the Multi-Spectral Imager (MSI).

    The imager provides information about aerosol and clouds in the horizontal direction. Therefore, it complements vertical information gained by the active instruments. In particular, the horizontal variability of atmospheric conditions and meso-scale cloud field structures in weather systems are supposed to be identified and characterized with MSI. In order to achieve that goal,

the imager swath is broader than the active instruments and broadband radiometer (BBR) footprints (Wehr, 2006). MSI's swath is 150 km. It is tilted away from sun-glint affected regions. This results in roughly 35 km of the swath being situated west of nadir and roughly 115 km east of nadir taking into account EarthCARE's afternoon orbit. In the past, a non-compliance of the MSI visible - near-infrared - shortwave infra-red (VNS) camera's visible (VIS), near-infrared (NIR) and shortwave-infrared 1 and 2 (SWIR1 and SWIR2, respectively) channels have been noticed regarding a spectral central wavelength shift as a

function of across track pixel or viewing angle (Wehr et al., 2023). This Spectral MIsaLignment Effect (SMILE) is caused by imperfections in the band-pass filters on the curved optical lenses (e.g., Wehr et al., 2023; Wang et al., 2023). Consequently, mitigation strategies have been implemented by ESA and industry. A lens barrel rotation for the VIS channel and a full re-design of the NIR channel have been accomplished. However, no efficient technical solution was found for the SWIR1 and SWIR2 channels. Even though the VIS channel offers a more consistent central wavelength (CWVL) close to nadir now,

there still is a significant shift present when the entire across track dimension is considered. Other imagers, like the Medium Resolution Imaging Spectrometer (MERIS), Hyperion or OLCI (Ocean and Land Colour Instrument) also had to handle spatial misalignments of the wavelength and different correction schemes have been developed (e.g., Bourg et al., 2008; Dadon et al., 2010; Kritten et al., 2020). Before any SMILE correction methodology can be developed, it is important to quantify the SMILE effect itself. Wang et al. (2023) already studied the influence of MSI's SMILE on cloud retrievals for shallow warm and deep

convective clouds over ocean surfaces and found a negligible impact, judging from an associated error typically within 10%. We will pick up where this study has started. However, our main focus here will be on smaller optical thicknesses (OT) of not only clouds, but also aerosols, over land surfaces. A higher impact and larger errors are to be expected due to the increasing spectral influence of these surface types. Therefore, the spectral response functions, the resulting CWVL for MSI VNS bands and the error metrics used in this study will be presented in section 2. Further, we assess the influence of the SMILE on the forward

models used for the European Level 2A MSI aerosol and cloud algorithms and their underlying assumptions quantitatively in section 3. In section 4, we additionally investigate the SMILE impact on the corresponding products of aerosol optical thickness (M-AOT, Docter et al., 2023) and cloud optical and physical properties (M-COP, Hünerbein et al., 2023a). Finally, a summary and a conclusion of the results and an outlook for the respective products will be given in section 5.

## 2   MSI's Spectral MIs-aLignment Effect (SMILE)

MSI's VNS camera has four bands. The VNS channels are needed for cloud detection, cloud type, cloud phase, cloud optical and micro-physical properties, scene identification (Hünerbein et al., 2023a, b) as well as aerosol properties (Docter et al., 2023) and for the surface reflectance estimation in a non-cloudy atmosphere.





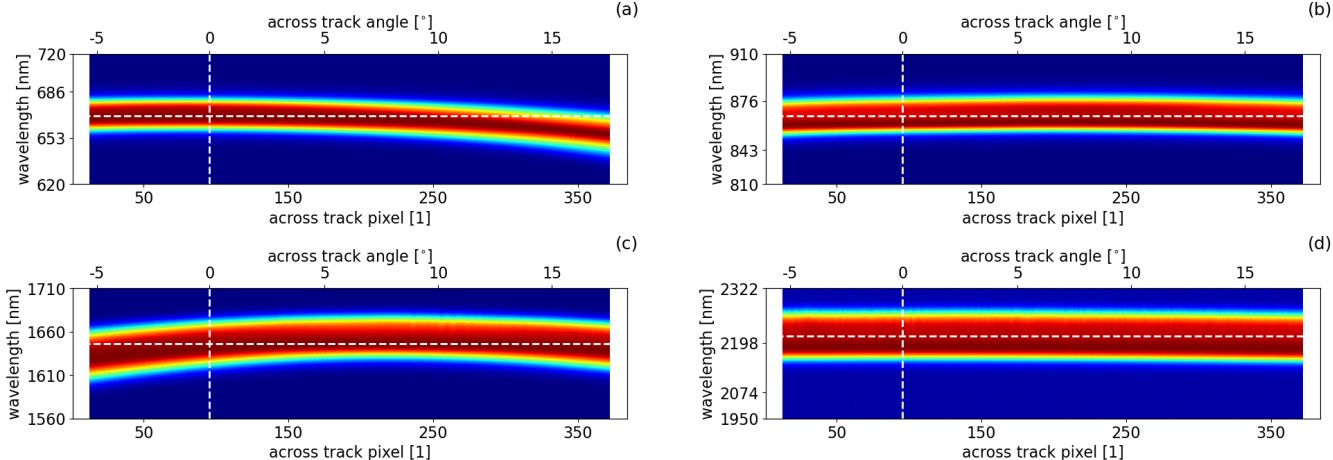

**Figure 1.** MSI response functions in dependence of across track pixel (bottom x-axis) or across track angle (top x-axis) and wavelength (y-axsis) for each MSI band: VIS (a), NIR (b), SWIR1 (c), SWIR2 (d). White dashed vertical lines indicate the nadir pixel and white dashed horizontal lines represent the corresponding central wavelength.

Spectral response functions are available for each of MSI's bands and for each across track pixel (Fig. 1). Additionally, the associated across track angle is shown. This angle is based on MSI pointing information, which is also available for each across track pixel. In particular, the across track angle is based on the viewing elevation angle in the satellite coordinate system converted to viewing zenith angle first. However, here, negative viewing zenith angles are used for west of nadir pixel and positive angle values are used east of nadir in order to distinguish both sides. Consequently, for the purpose of avoiding any confusion of negative zenith angle values with measurements from ground, we are calling this angle across track angle instead of viewing zenith angle from now on. In particular, the VIS and the SWIR1 channel show a strong shift of the response function center across track when compared to nadir.

The corresponding CWVLs (Fig. 2) can be calculated based on these response functions ($\eta_{p,\lambda}$) for each across track pixel ($p$) with:

$$\lambda_{c,p} = \frac{\int_{\lambda_1}^{\lambda_n} \lambda \cdot \eta_{p,\lambda} \, d\lambda}{\int_{\lambda_1}^{\lambda_n} \eta_{p,\lambda} \, d\lambda} \tag{1}$$

The integration bounds considering wavelength $\lambda$, that are used for the respective bands, are $\lambda_1$=600 nm and $\lambda_n$=750 nm for the VIS band, $\lambda_1$=790 nm and $\lambda_n$=930 nm for the NIR band, $\lambda_1$=1530 nm and $\lambda_n$=1770 nm for the SWIR1 band and $\lambda_1$=1800 nm and $\lambda_n$=2300 nm for the SWIR2 band.

In the following, only CWVLs which are based on smoothed response functions are used. This is mainly done in order to suppress noise present in the measured response functions of the SWIR2. The noise is caused by imperfections in the measuring set-up and is not expected to be physically present in the channel itself. While the choice of smoothed versus default response



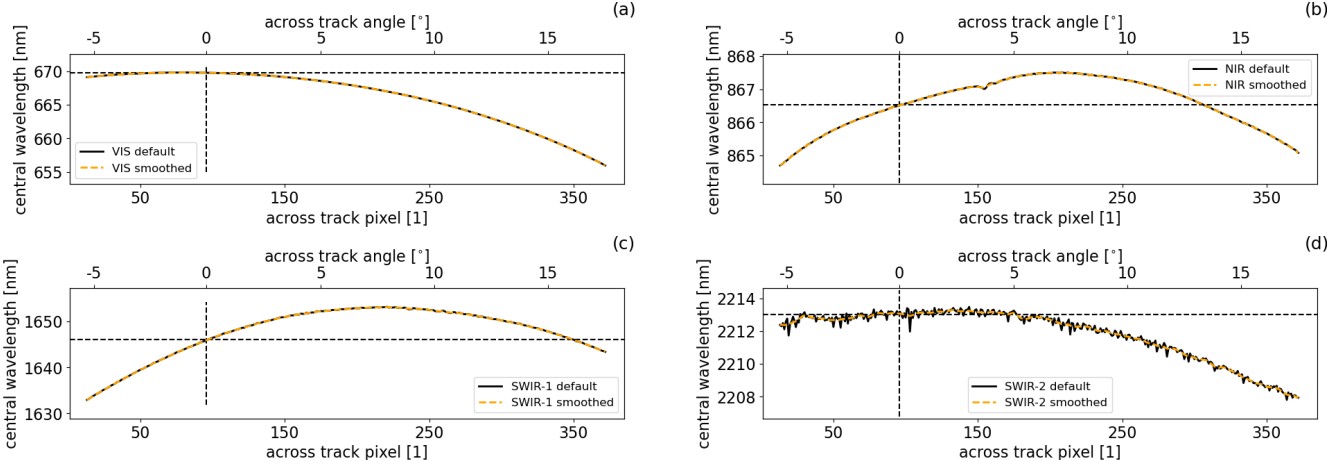

**Figure 2.** MSI CWVL in dependence of across track pixel (bottom x-axis) or across track angle (top x-axis) for each MSI band: VIS (a), NIR (b), SWIR1 (c), SWIR2 (d). Black lines show the CWVL based on the default response functions, while yellow lines show CWVL for smoothed response functions.

**Table 1.** Across track central wavelength variation for the four VNS bands of EarthCARE's MSI

|  | VIS | NIR | SWIR1 | SWIR2 |
|---|---|---|---|---|
| Nominal centre wavelength in nm (Wehr et al., 2023) | 670 | 865 | 1650 | 2210 |
| Central wavelength at nadir in nm | 669.8 | 866.5 | 1646.2 | 2213.0 |
| Total, absolute min/max variation across track in nm | 13.8 | 2.8 | 20.2 | 5.3 |
| Variation of -/+25 km (west/east) around nadir in nm | -0.1/-0.6 | -0.8/+0.5 | -7.2/+4.8 | -0.3/0.2 |
| Total, absolute variation +/-25 km around nadir in nm | 0.5 | 1.4 | 12.0 | 0.5 |

functions has no significant impact on the VIS, NIR, and SWIR1 CWVL estimates, it enhances the CWVL across track shape for the SWIR2 band.

The CWVL variation (Tab. 1) is about 3 nm and 5 nm for the NIR and SWIR2 channel, respectively. However, it exceeds 10 nm for the VIS and SWIR1 channel if the entire across track dimension is considered. The corresponding total CWVL variation is about 14 nm for the VIS band and about 20 nm for the SWIR1 band. Considering only the nadir region +/-50 pixel (corresponding to approximately +/-25 km), this CWVL variation decreases for all bands.

## 3 Impact of the SMILE on the MSI aerosol and cloud retrieval assumptions

Variations of the spectral response functions and CWVLs of MSI bands have an impact on the measurement accuracy itself and will impose an additional uncertainty in the respective retrieval algorithms if not accounted for or corrected beforehand.





In particular, the impact of not accounting for the SMILE in the forward models on which M-AOT and M-COP are relying is investigated in the following. Therefore, we analyze the errors that are introduced due to only using the nadir spectral response function or nadir CWVL ($nadir$) instead of the ones corresponding to an individual across track pixel ($true$). We quantify the uncertainty introduced due to the SMILE using the relative error ($\delta x$):

$$\delta x = \frac{x_{nadir} - x_{true}}{x_{true}} \tag{2}$$

First, we analyze the impact on the forward model assumptions considering spectral gas absorption, surface albedo as well as on aerosol and cloud optical properties in section 3.1. Secondly, we quantify the errors introduced on the forward simulated normalized radiance in section 3.2 for each band individually.

### 3.1 Impact on spectral forward simulation inputs

#### 3.1.1 Impact on gas absorption description

Gas concentrations, temperature and pressure of the US standard atmosphere (Anderson et al., 1986) have been used to extract high resolution absorption cross sections from the CKDMIP (Correlated K-Distribution Model Intercomparison Project, Hogan and Matricardi (2020)) database. These then have been used to calculate highly resolved transmissions which are convolved with the spectral response functions of MSI. Figure 3 shows the gaseous transmissions for all four MSI VNS channels. Bands which have quite a strong shift of CWVL in the across track dimension, i.e. the VIS and SWIR1 bands, are also shifting towards stronger gas absorption features. In particular, the VIS channel is affected by the absorption of ozone and water vapour in general. The shift of the center of the response function towards shorter wavelength for the most east across track pixel exposes these pixel to higher water vapour and ozone absorption than what would be expected for the nadir pixel. Considering the SWIR1 band, the across track shift from west to east will expose a pixel to less methane absorption than at nadir in the first place. However, considering across track pixel in the most eastern part of the track, their response functions are then not only shifting out of methane absorption, but are partly shifting into the carbon dioxide absorption feature at about 1595 to 1615 nm.

In order to quantify the error introduced by the SMILE on gas transmission assumptions, eq. 2 is applied on these transmissions. The resulting impact can be seen in figure 4. While the relative error is below +/-0.5% for the NIR, SWIR1 and SWIR2 band, it is 3.3% for the VIS channel for the outermost across track pixel on the east part of the swath.

As a consequence of the underestimation of absorption in that part of the track for the VIS band, an ordinary atmospheric correction scheme would produce underestimated top-of-atmosphere (TOA) signals corrected for gaseous absorption. The Level 2A algorithms for cloud optical and physical properties as well as aerosol optical thickness apply such atmospheric correction schemes, which take the absorption by atmospheric gases in MSI bands into account. For the retrievals, it will be crucial to correct for gaseous absorption accurately since, for example, optical thickness due to gases alone for nadir VIS is about 0.07 for the US standard atmosphere. This in turn is already in the order of expected common background aerosol optical thickness. Due to the shift, the error in gas optical thickness in the VIS band can be up to -0.03 across track. This uncertainty alone could potentially already lead to a failure of the absolute accuracy requirement for aerosol optical thickness if the SMILE is not taken into account for spectral gas assumptions.




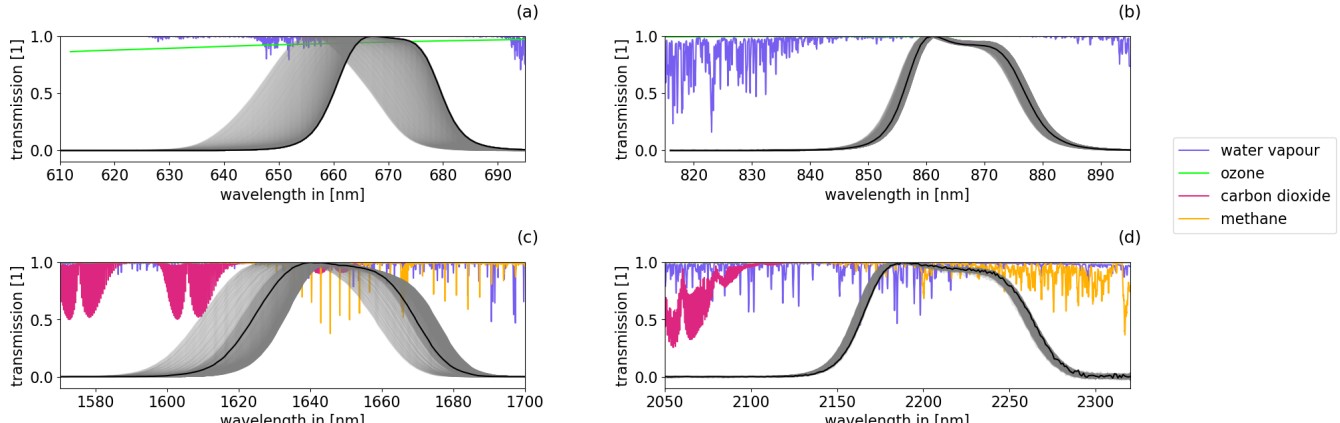

**Figure 3.** Gas transmission for MSI VIS (a), NIR (b), SWIR1 (c) and SWIR2 (d) channel. Grey lines indicate MSI spectral response functions across track, black lines indicate MSI nadir filter function and colored lines represent different gases of the atmosphere.

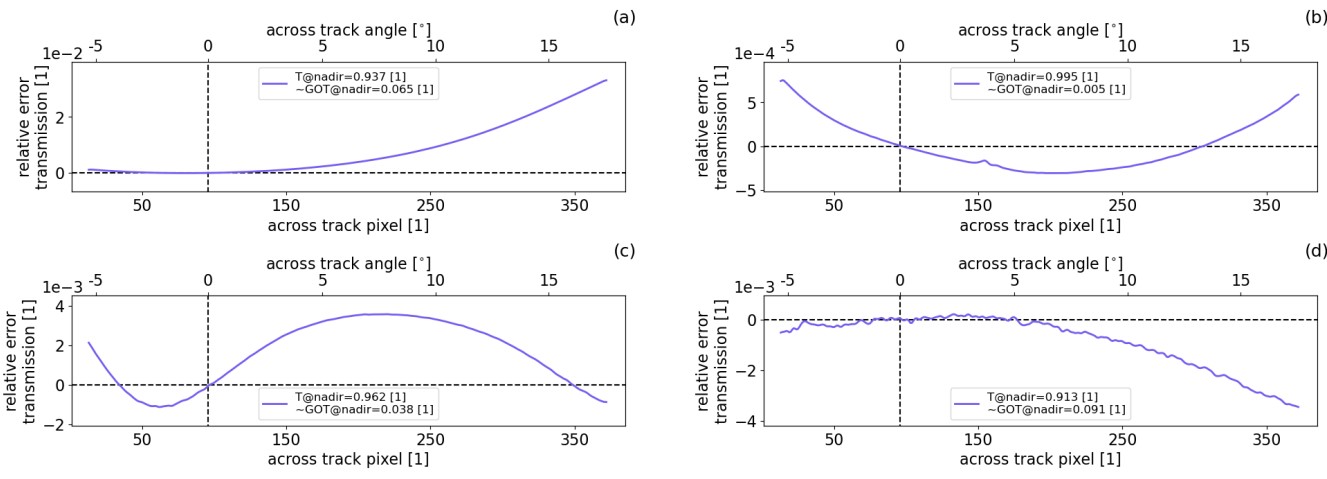

**Figure 4.** Relative error of convolved transmission when nadir central wavelength would be assumed across track instead of true central wavelength based on spectral response functions in dependence of across track pixel (bottom x-axis) or across track angle (top x-axis) for MSI VIS (a), NIR (b), SWIR1 (c) and SWIR2 (d) channel. Dashed black vertical lines indicate nadir.



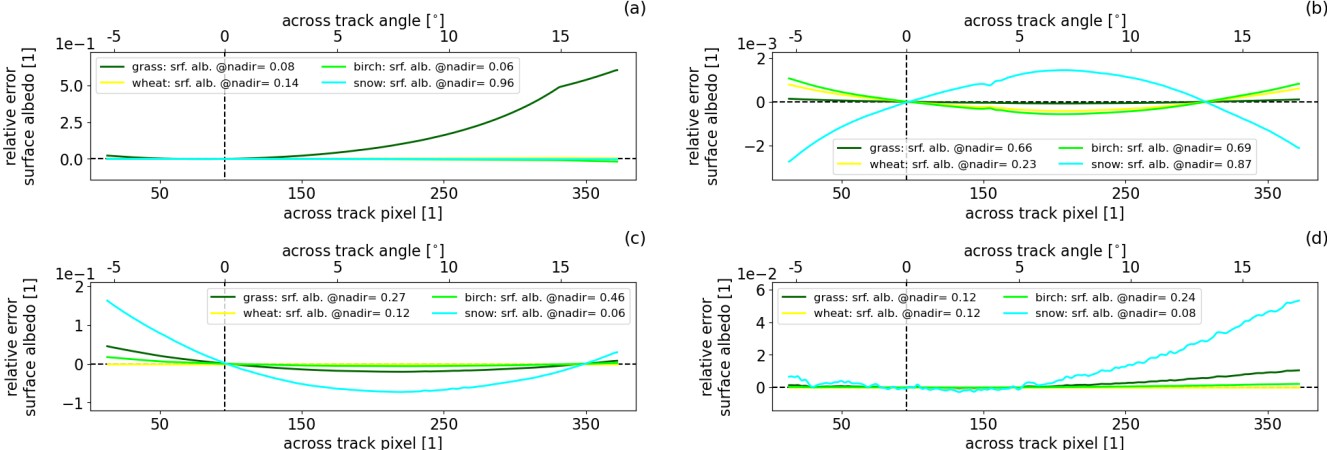

**Figure 5.** Same as Fig. 4, but for surface reflectance. Colored lines indicate different surface types.

### 3.1.2 Impact on the land surface description

Surface reflectances for different land surface types are used to quantify the error as a consequence of neglect of the SMILE. Grass, wheat and birch reflectance are taken from Bowker et al. (1985). Snow (medium grained) reflectance is taken from the

ECOSTRESS spectral library version 1.0 (Meerdink et al., 2019; Baldridge et al., 2009). The corresponding errors when only the nadir CWVL would be assumed instead of the actual across track CWVL are summarized in figure 5. The VIS channel is located close to the red edge of green vegetation surface reflectance. Therefore, small central wavelength shifts can lead to uncertainties due to the rapid change in surface reflectance that are characteristic for these spectra. The nadir central wavelength is situated closer to the red edge than the central wavelength corresponding to viewing zenith angles towards +15° causing the

relative error (Fig. 5a) in surface reflectance to vary between -0.8% (birch) and 50% (grass) depending on the spectral signature of the vegetation itself. For cloud (clouds with an optical thickness lower 10) and aerosol retrievals an overestimation of surface reflectance alone would lead to an underestimation of the cloud or aerosol optical thickness over vegetated land. This is due to an overestimation of the surface reflectance towards shorter central wavelength causing a lower part of the TOA signal to be associated to cloud or aerosol in the VIS channel. The NIR channel has low relative errors (Fig. 5b) of surface reflectance when

assuming the nadir reflectance instead of the true central wavelength reflectance. They are most pronounced for a snow surface and vary between -0.19% and 0.13%. The relative error of surface reflectance in the SWIR1 channel (Fig. 5c) expresses the same behaviour for snow, grass and birch surfaces. Towards shorter wavelength than the nadir central wavelength, the surface reflectance is overestimated by up to 16.3%, 4.5% and 1.7%, respectively. An underestimation of surface reflectance of -7.3%, -2.1% and 0.6%, respectively, can be found towards longer wavelengths than nadir. The SWIR2 shows relative errors (Fig. 5c)

of up to about 1% for grass and up to 5.3% for a snow surface.

The cloud detection and type retrieval is based on the combination of VIS, NIR and SWIR signals, e.g. the ratio of VIS-to-NIR. The thresholds rely on the assumption that the spectral signatures of cloud-free pixels and pixels covered by different





cloud types differ. Consequently, each threshold test is dedicated to a certain surface classification and cloud type as the measured TOA signal of a cloud with an optical thickness lower than 10 includes a significant component from the surface,

which has to be characterized accurately. In general, uncertainties in surface reflectance will reduce the accuracy of the cloud mask, cloud and aerosol retrievals.

### 3.1.3 Impact on aerosol and cloud optical property descriptions

In order to investigate erroneous assumptions about aerosol and cloud optical properties in the respective LUTs used in the Level 2A retrieval, we are relying on the HETEAC (Hybrid-End-To-End Aerosol Classification, Wandinger et al., 2023) model

for EarthCARE for aerosols and on the description of Baum et al. (2014) for ice clouds and spherical water droplet for liquid clouds. The effective radius that is associated with these two cloud types and presented in the following is 10 and 5 μm, respectively.

Figures 6, 7 and 8 show relative errors in optical property assumptions, i.e. extinction, single scattering albedo and scattering phase function, respectively. As to be expected based on each channels response function and the spectral behaviour of each

individual quantity, the most affected channels are VIS and SWIR1 across all quantities and scatterer types.

Aerosol optical properties are prone to erroneous assumptions if not spectrally resolved in M-AOT. In particular, aerosol in the VIS and SWIR-1 band need to be resolved. The M-AOT forward model would be exposed to optical thickness errors of up to -4% in the VIS (Fig. 6a) and -3% to 2% in SWIR-1 channel (Fig. 6c) for both HETEAC fine mode aerosol types, if aerosol extinction in these bands is not properly spectrally resolved for radiative transfer simulations. Theoretically, such an

underestimation in spectral AOT, as for the VIS channel, would result in an overestimation of retrieved AOT if only the nadir central wavelength would be used in Look-Up Tables (LUT), since the TOA radiance that is forward simulated for a given optical thickness will be lower than the true TOA radiance that would be measured.

While, relative errors in the single scattering albedo (Fig. 7) of HETEAC dust and fine mode strong absorbing aerosol are also most prominent in these two bands (VIS and SWIR1), they hardly reach 1%. Nonetheless, relative errors of the scattering

phase function (Fig. 8) used for radiative transfer simulations could reach up to 2.3 to 2.7% for fine mode aerosol and 4 to 5% for coarse mode aerosol in the VIS and still 1.4% in the SWIR-1 band for sea salt, if one would ignore the spectral dependency within the bands.

In contrast, only the SWIR1 and SWIR2 bands, which are the water and ice absorption channels, exhibit a slight sensitivity to the spectral across track variation in the single scattering albedo, as to be theoretically expected based on the spectral

dependency of the imaginary parts of water and ice. Further the water and ice absorption channels (SWIR-1 and SWIR-2) are primarily a function of cloud particle size, this is reflected in the results. The relative error for the single scattering albedo getting stronger for larger cloud particle (not shown here). Nonetheless, relative errors hardly ever reach even 0.1%. In Figure 8, as an example, the relative error of the phase function are given for water cloud with an effective radius of 5 μm. The phase function for VIS and SWIR-1 bands are most effected a relative error up to 3% are found.



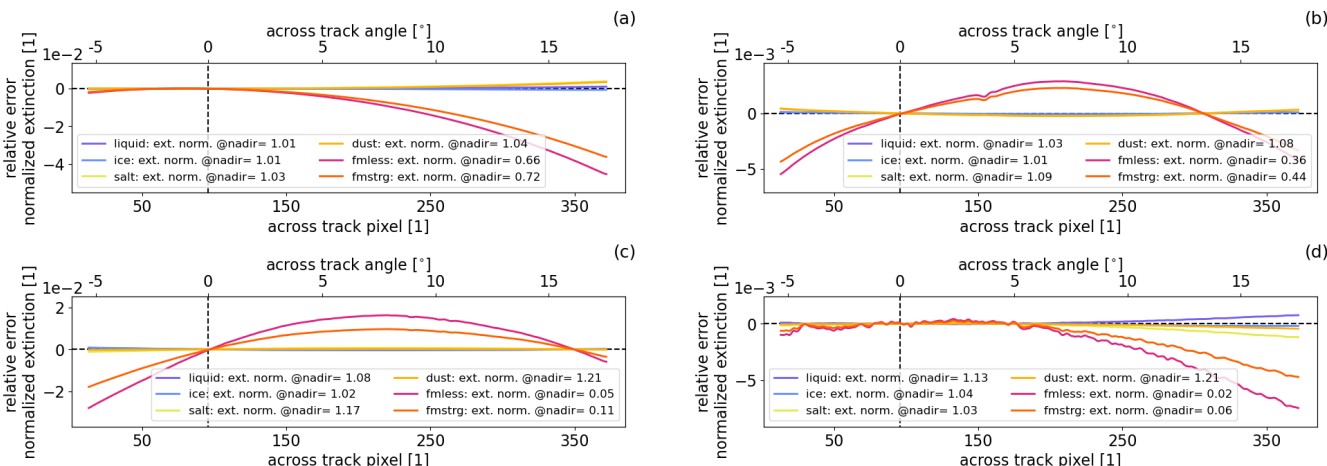

**Figure 6.** Same as Fig. 4, but for the relative error of extinction. Colored lines indicate different scatter types.

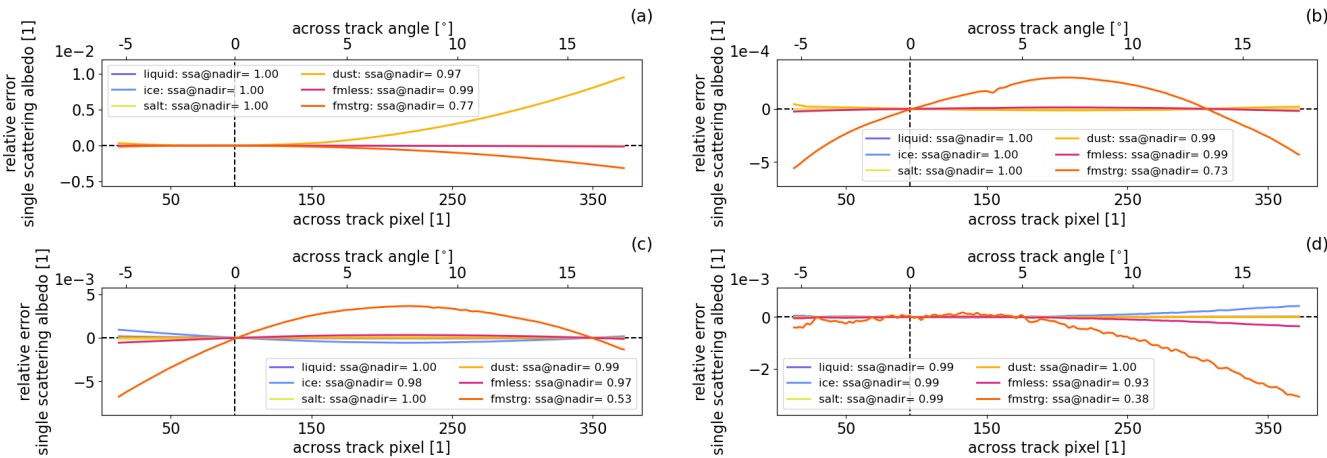

**Figure 7.** Same as Fig. 6, but for the relative error of the single scattering albedo.





**Figure 8.** Relative error of scattering phase functions when nadir central wavelength would be assumed across track instead of true central wavelength based on spectral response functions for MSI VIS (first column), NIR (second column), SWIR1 (third column) and SWIR2 (fourth column) channel. Rows are corresponding to different scatter types: liquid cloud with an effective radius of 5 μm (first row), ice cloud with an effective radius of 10 μm (second row), HETEAC sea salt (third row), HETEAC non-spherical dust (fourth row), HETEAC fine mode less absorbing aerosol (fifth row) and fine mode strong absorbing aerosol (sixth row).





## 3.2 Effect on aerosol and cloud TOA normalized radiances

Radiative transfer simulations have been carried out using the Doubling-Adding KNMI (DAK; de Haan et al., 1987; Stammes et al., 1989; Stammes, 2001) for cloud considerations and the Matrix Operator Model (MOMO; Fell and Fischer, 2001; Hollstein and Fischer, 2012) for aerosol investigations in order to better understand:

- How are these assumption errors interacting?

- How are they affecting the forward simulated TOA signal in M-AOT and M-CLD if the SMILE would be neglected there?

Each of MSI's bands has been resolved in one nanometer steps between minimum and maximum central wavelength of each band. This leads to 16 simulations for the VIS (655-670 nm), five simulations for the NIR (864-868 nm), 23 simulations for the SWIR1 (1632-1654 nm) and 8 simulations for the SWIR2 (2207-2214 nm) band. These spectrally resolved simulations can be used to quantify errors for different cloud and aerosol settings, if one would stick to the heritage single central wavelength per band setting as used in current MSI Level 2A processors. For the following quantification, we will rely on the normalized TOA radiance, which is defined as the ratio of spectral radiance to spectral irradiance. Further, it has not been accounted for gas absorption here, since the respective level2 aerosol and cloud retrieval are correcting MSI Level 1 measurments outside of the respective forward models.

Exemplary, Fig. 9 shows the relative error of normalized radiance for each of the four HETEAC aerosol types with an AOT(550 nm) of 0.3, one liquid cloud (effective radius is $5\,\mu m$) and one ice cloud (effective radius is $10\,\mu m$) with a COT of 10 for one observation geometry. The surface type chosen for aerosol cases is grass relying on surface albedo values as used for relative error calculations in Fig. 5. For cloud cases, the influence of the surface has been neglected due to higher optical thicknesses used for these scenarios and is 0.05.

Whenever optical thickness is rather low, as for aerosol cases, and the influence of the spectral variation of the surface becomes non-negligible within a band, the relative error becomes larger. As expected, based on relative errors present for the optical properties and the surface albedo, the VIS band is most sensitive to any omission of the SMILE, while the effect is insignificant for the NIR band.

The relative error of TOA normalized radiance for aerosol cases slightly increases with decreasing particle size for the VIS (Fig. 9a), SWIR1 (Fig. 9c) and SWIR2 (Fig. 9d) bands. E.g., the relative error reaches up to 27% for corase mode (CM) and 30% for fine mode (FM) aerosols for across track angles larger than 15° for the VIS band. Relative errors of TOA normalized radiance closely resemble surface albedo errors and vary between -1.9% (CM) to -2% (FM) and 4.2% (CM) to 4.4 (FM) for the SWIR1 and can still reach up to 0.9% (CM) and 1% (FM) for the SWIR-2 band. The relative error is most pronounced for ice clouds. The VIS band exhibit errors up to 2% with an across track angle greater than 10° (Fig.9a). The SWIR-1 band shows a significant error over the whole across track dimension (1.4 to -0.8% for ice, 0.4 to -0.2% for water). As the combination of VIS/SWIR-1 are the base to retrieve COT and effective radius, the finding has to be considered. The relative error of the NIR and SWIR-2 are less than 0.1% and 0.05%, respectively.





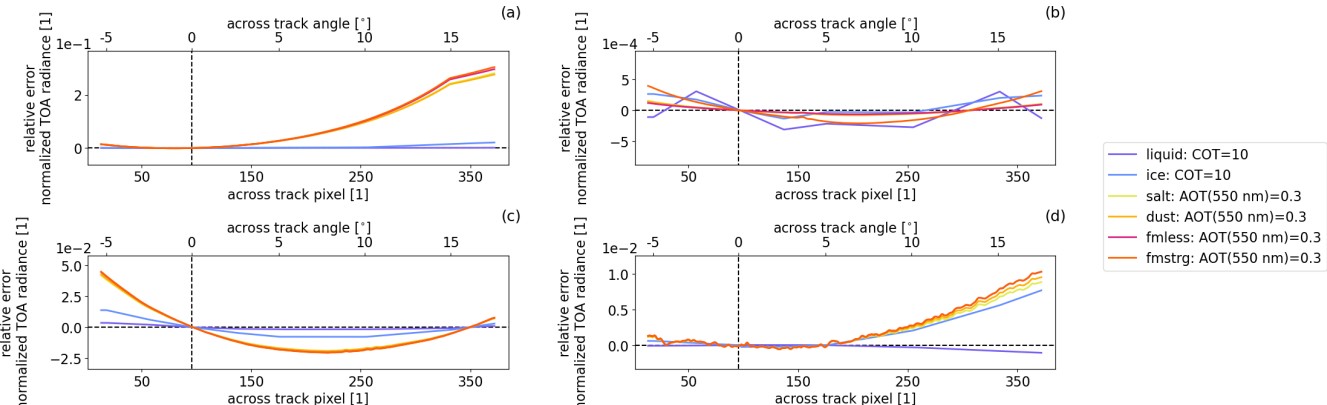

**Figure 9.** Same as Fig. 6, but for the relative error of the normalized TOA radiance over a grass surface. Sun zenith angle is $40°$, relative azimuth difference is $130°$, COT is 10 and AOT(550 nm) is 0.3. Colored lines show different scatter types.

## 4 Error quantification for MSI L2 aerosol and cloud products

### 4.1 Synthetic SMILE scene

To quantify errors introduced into the Level 2A retrievals M-CLD and M-AOT due to the SMILE of MSI, the European EarthCARE simulator (ECSIM, Donovan et al., 2023) sub-module for MSI is used to generate an MSI SMILE block scene. Therefore, a first set of artificial sub-cases has been defined within one EarthCARE-MSI frame. Seven aerosol cases are present over ocean and land surfaces, i.e. over evergreen broad-leaf forest. Additionally, two cloud types (stratocumulus and cirrus) are defined with two different cloud effective radii ($40\,\mu m$ for cirrus, $6\,\mu m$ for the water cloud ) and two cloud optical thicknesses

(1 for cirrus, 10 for the water cloud) over land (Fig. 10), which is divided in two different surface types: green vegetation and barren to sparsely vegetated land. Cloud and aerosol quantities are altering along track and stay constant across track. Atmospheric conditions are constant regarding skin temperature ($293.8\,K$), surface pressure ($1007.14\,hPa$), horizontal wind speed ($4\,ms^{-1}$) and total column gas amounts (water vapour: $29.3\,kgm^{-2}$, ozone: $0.006\,kgm^{-2}$, carbon dioxide: $5.2\,kgm^{-2}$). The scene, that includes a limited subset of real-world conditions, can be used to do a first investigation of uncertainties

introduced solely due to the SMILE if M-CLD and M-AOT processors are not accounting for it.

Two ECSIM runs have been carried out for this artificial subset of an EarthCARE MSI frame. The first run assumed ideal conditions that correspond to what would be observed if the across-track response functions all corresponded exactly to the nadir one. The other run contains the SMILE affected quantities for which the SMILE has been implemented in the model radiative transfer. Therefore, surface, gas, aerosol and cloud properties, Rayleigh and solar in-band irradiance have been spec-

trally resolved. In order to use these two scenes in M-AOT and M-CLD processors, surface and particle types have been prescribed for this artificial scene as defined in Fig. 10. This is due to the dependence of M-AOT and M-CLD on background climatologies of e.g. surface setting or aerosol type over land.





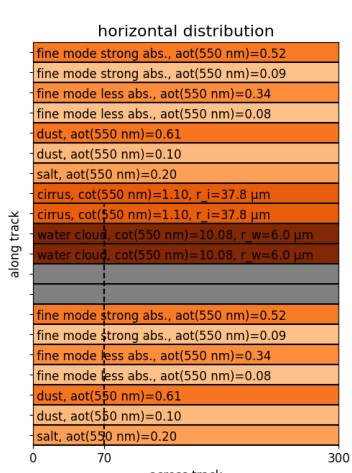 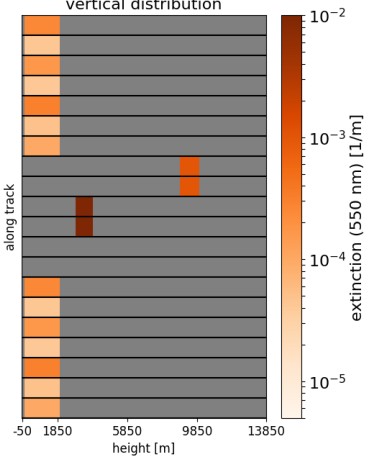 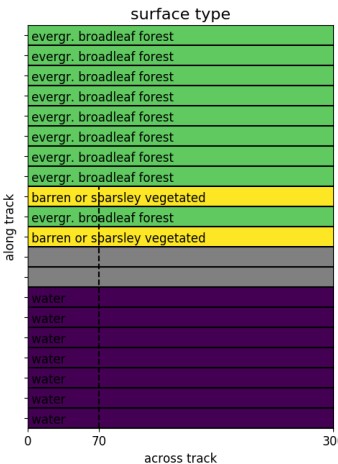

**Figure 10.** Horizontal (a) and vertical (b) distribution of aerosols and clouds and their classification within the respective stripes along with the surface distribution (c) in the synthetic MSI SMILE scene. Dashed black lines in (a) and (c) indicate nadir.

Resulting differences between the two runs are once again shown using the relative error of normalized TOA radiances in Fig. 11 over land and Fig. 12 over ocean. Some residual noise is present in the simulations of the synthetic scenes which is caused by numerical inaccuracies in the very highly resolved spectral calculations.

As to be expected, strongest effects to the SMILE are present for the VIS and SWIR1 bands. When comparing the VIS channel over land (Fig. 11a) and water (Fig. 12a), it can be seen that the chosen spectral surface characteristics over land decrease the relative error in normalized TOA radiance. Over ocean, an relative error of up to -6.7% is present for fine mode aerosols with a low loading. Over land, the relative error in the TOA normalized radiance of the VIS channel only reaches -2% comparing the MSI smile scenes for aerosols. This is due to a relatively flat spectral behaviour of the forest example. Relative errors of the bidirectional reflectance distribution function (not shown) used for the synthetic scene creation only reach up to 1.5% for the VIS band. However, since the synthetic scene includes absorption due to gases, the overestimated water vapour and ozone absorption towards across track pixel in the east of the swath would lead to an overall underestimation in TOA normalized radiance there. The cirrus is mainly influenced by the surface and gas absorption as the aerosol and follow the behaviour for the same reason. In contrast, the relative error of cirrus over barren land is up to 1.3%, which comes from the different spectral behavior of this surface type. The optical thicker clouds, as the stratocumulus clouds, the effects due to the spectral variation are hardly noticeable.

For the SWIR-1 channel, the relative error of TOA normalized radiance can vary between -2% (west of nadir) to 1.1% (east of nadir at an across track angle of about 7.5°) over ocean (Fig. 12c) for fine mode aerosols. Over land, relative errors of TOA normalized radiance can vary between -0.3% (east of nadir at an across track angle of about 7.5°, cirrus over sparsely vegetated land) and 1.2% (west of nadir, cirrus over forest) for clouds. Here, the cloud properties play the main role as the





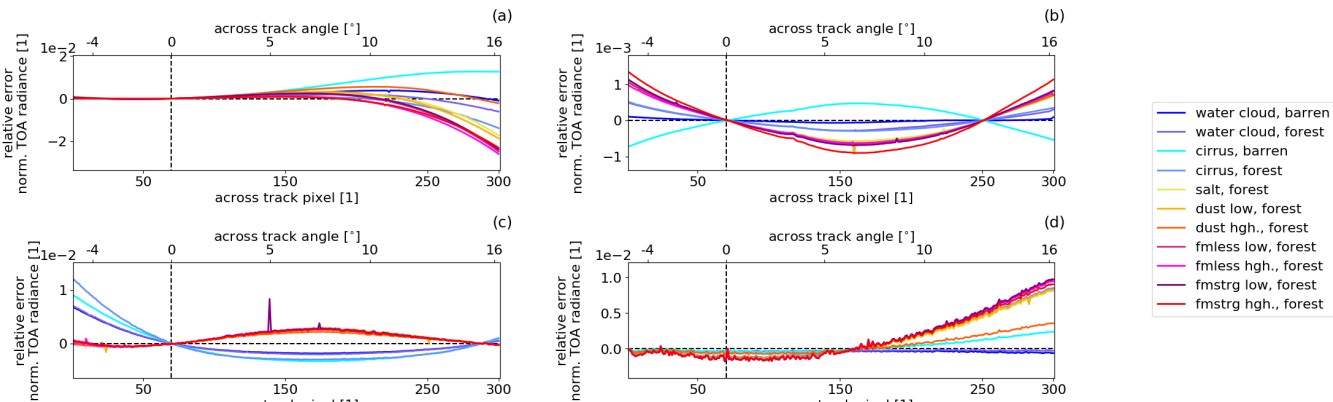

**Figure 11.** Relative error of TOA normalized radiance in dependence of across track pixel (bottom x-axis) or across track angle (top x-axis) for VIS (a), NIR (b), SWIR1 (c) and SWIR2 (d) channel over land in the synthetic MSI SMILE scene. Dashed black vertical lines indicate nadir.

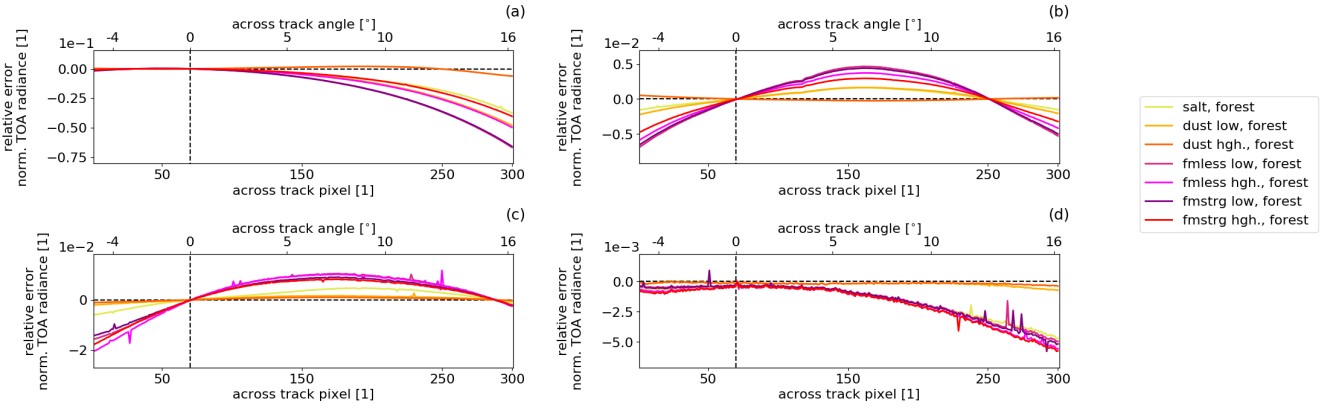

**Figure 12.** Same as Fig. 11, but over ocean and for aerosol only.

water and ice absorption in SWIR1 are primarily a function of the cloud particle size, this is reflected in the results. The effects of the SMILE for the NIR and SWIR-2 channel are below +/-1% over ocean (Fig. 12b, d) and land (Fig. 11b, d).

## 4.2 Error implication for MSI aerosol and cloud optical thickness

240 Using the two synthetic scenes in the cloud and aerosol algorithms, that have not yet been mitigated to account for the SMILE, allows to directly compare M-COP and M-AOT output products. Consequently, this enables the quantification of errors solely introduced due to the SMILE if neither Level 2A processor would account for the SMILE, nor Level 1 would be corrected.

Fig. 13 shows the relative error of COT and AOT at 670 nm over land and ocean. The relative error of retrieved AOT at 670 nm is highest (-48%) over land (Fig. 13a) east of nadir for fine mode strong absorbing aerosol with a low aerosol loading. The





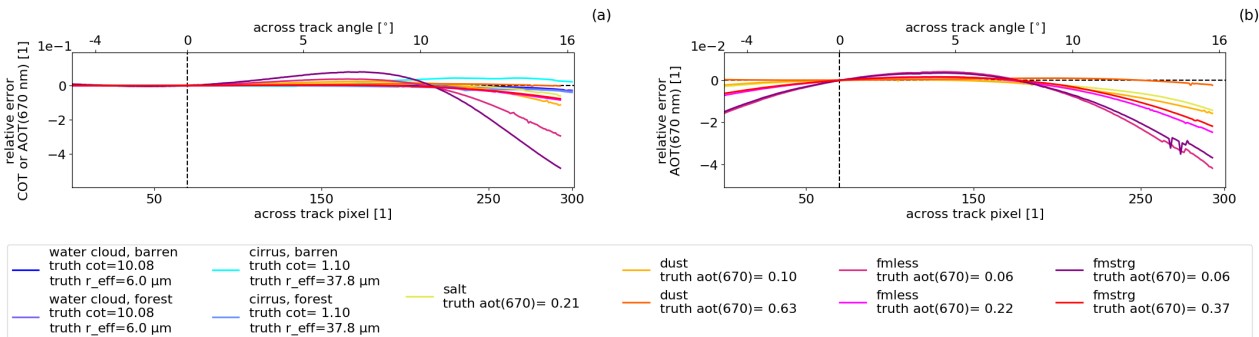

**Figure 13.** Relative error of M-COP COT and M-AOT AOT at 670 nm over land (a) and M-AOT-AOT at 670 nm over ocean (b) in dependence of across track pixel (bottom x-axis) or across track angle (top x-axis) for the MSI SMILE scene. Dashed black vertical lines in (a) and (b) indicate nadir.

error decreases for higher AOT loading and bigger particles such as dust or salt aerosols over land. Therefore, AOT at 670 nm over land can be underestimated by up to 0.03 for the fine mode less absorbing, forest example with an actual AOT of 0.06.

Over ocean, the general behaviour of the relative error of aerosol optical thickness at 670 nm (Fig. 13b) appears to be similar to the one over land, even though the absolute values are lower reaching up to only -4%.

The cloud optical and physical properties retrieval (M-COP) are based on the VIS, SWIR1 and TIR bands. COT is mainly a function of the VIS band and the cloud effective radius of the SWIR1 band. The relative error for COT is dominated by the surface properties for the optical thin cirrus, but also the optical thicker stratocumulus cloud is affected by the SMILE by up to -3% (Fig. 13a). The relative error of the cloud effective radius with a small effective radius (6 μm) are low, but with a higher cloud effective radius the results show a pronounced relative error from 7.5% up to -5% (not shown, curve follows Fig. 11c). The results are reasonable, but as the uncertainty of effective radius for optical thin clouds are in general very high this value has to be taken with caution.

## 5 Conclusions and Outlook

Passive imager measurements of MSI on EarthCARE and are needed for additional knowledge about the horizontal cloud and aerosol distribution in addition to active measurements that provide information about the vertical distribution. In order to achieve good quality of Level 2A aerosol and cloud products based on MSI measurements, a reasonable accuracy and a good characterization of the instrument is needed. A small SMILE is not an uncommon effect for imagers. However, MSI shows across track CWVL variations in the VIS and SWIR1 band of up to 14 and 20 nm, respectively. The main purpose of this study is the assessment of MSI's SMILE on the European Level 2A aerosol and cloud retrievals in order to establish a way forward in accounting or correcting for it in the retrieval procedures. Therefore, we investigated its impact on the forward models and on their underlying spectral assumptions (e.g. of gas absorption, surface reflectance and aerosol and cloud optical properties)





within each band. Additionally, we created an artificial MSI SMILE scene to directly assess the error introduced on M-AOT and M-COP products, if this effect would be ignored within the retrieval algorithms.

Already only considering the impact on gas absorption, the relative error is up to 3.3% in the VIS related to the absorption of ozone and water vapour. While over ocean surfaces, the effect of MSI's SMILE on the retrieved optical properties is less pronounced than over land, it still persists, in particular for low optical thickness and small particles. Errors there are about
1.4% for coarse mode aerosols to 5% for fine mode aerosols with a low AOT. Depending on the underlying surface and its associated spectral variation within the VIS band, relative errors in surface albedo can reach up to 50% for e.g. grass if any retrieval relying on such information would not account for the SMILE. Already smaller surface variations within MSI bands, e.g. the presented example over forest, can lead to an underestimation in AOT of up to -50% for fine mode aerosol with a low loading or an underestimation in COT of up to -4% for thin cirrus. However, such smaller variations in the spectral surface
reflectance within a band become less significant with increasing AOT as well as COT. Nonetheless, the overall error over land is expected to strongly vary depending on the actual underlying land surface type. E.g., since there are already larger absolute relative errors of TOA normalized radiance present for grass in the VIS band than for forest, also retrieved OT over grass is expected to show larger errors if the SMILE is not accounted for. Additionally, a reverse behaviour of relative error of TOA normalized radiance was found for the VIS band for grass and forest. This seems to indicate that the influence of
errors in the surface assumptions become more important with increasing variation in the spectral surface for the measured TOA signal. Hence, ultimately, this effect could also lead to an overestimation for surfaces like the grass example instead of an underestimation in OT as for the forest example, as long as the present OT is low.

Even using this basic approach to quantify errors induced due to SMILE neglection, our results show that the relative error over land for optical thin OT cannot be ignored. In order to avoid errors introduced due to the SMILE independent of its
actual magnitude and in order to allow for a more easily interpretation of the Level 2A products, any future developments or improvements to already existing algorithms should take this effect into account. In theory, there are two potential approaches to mitigate already existing retrieval algorithms: a correction of the measured Level 1 signal, following e.g. Bourg et al. (2008), Dadon et al. (2010), Kritten et al. (2020) or Jänicke et al. (2023), or accounting for the smile directly in the Level 2A retrieval procedures. The latter approach is identified as the one to be used for M-AOT and M-CLD product algorithms.

With the knowledge gained by this study, we are planning to adapt the auxiliary input data of cloud and aerosol look up tables (LUT), gas correction coefficients and surface parameterization coefficients to account for varying wavelength in the MSI Level 2A cloud and aerosol retrievals. This will be accomplished by re-generating the auxiliary data. In particular, the spectral variation of central wavelength will be represented in the across track angle dimension, which replaces the viewing zenith angle dimension in the respective LUTs. In fact, every viewing direction has an analogous across track angle direction
and an individual instrument response. Therefore, the number of LUT dimensions will not increase. Currently, all these changes to the existing MSI Level 2A processors are work in progress. While the mitigation to account for the SMILE is aimed to be ready before the launch of EarthCARE in 2024, still close attention should be paid to the SMILE when exploiting, interpreting and validating Level 2A data, in particular during the commissions phase.



*Author contributions.*  The manuscript was prepared by ND and AH. ND, RP and AH conceptualized the central wavelength calculation and
analysis. DD generated the SMILE scene simulations. Gracious advices and helps from JFM and PS for the DAK simulation. JF provided
the theoretical ground of the MOMO aerosol simulations. ME extracted the spectral response functions of MSI from ESA's Characterization
and Calibration Data Base (CCDB) and provided MSI pointing information file. All authors were involved in discussions and contributed
material and/or text to the manuscript.

*Competing interests.*  At least one of the (co-)authors is a member of the editorial board of Atmospheric Measurement Techniques.

*Acknowledgements.*  This work has been funded by ESA grants 4000112018/14/NL/CT (APRIL) and 4000134661/21/NL/AD (CARDI-
NAL). The authors thank Tobias Wehr for his support over many years and the EarthCARE developer teams for valuable discussions in
various meetings.





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
