# Peer review of "Assessment of the Spectral MIsaLignment Effect (SMILE) on EarthCARE's Multi-spectral imager aerosol and cloud property retrievals"

_EGUsphere, 2023_

## Author Comment (AC1)

**Reply to Anonymous Referee #1**

We thank the reviewer for their helpful and constructive comments. Please find our answers (black text) to the comments (blue text) in-line below. Respective changes are indicated in the revised manuscript in blue and are stated here in addition when a reviewer's comment leads to a substantial modification of the manuscript along with the updated line number, where necessary.

**General comments:**

This manuscript presents an investigation of the spectral misalignment Effect (SMILE) on aerosol and cloud retrievals with a focus over land. A number of potentially important sources of biases are assessed. The paper is well suitable to the scope of AMT, in terms of the question proposed to address.

I suggest major revision, considering the limitations and potential improvements as follows.

Writing could be improved. There are quite some grammar mistakes. I provided corrections for some of them, but there are more to be corrected. Please remove the words that were repeatedly used but unnecessarily or incorrectly, such as "already", 'e.g.', 'i.e.', and "therefore"(used when there was no causal relationship). Also, the tense was misused sometimes. A thorough examination must be performed while authors revise the manuscript.

Thank you for your suggestions. They have been included. In particular, we removed "already", "therefore" (except two), one "i.e." and several "e.g.". Additionally, we thoroughly checked the grammar after all reviewer comments had been accounted for in the updated manuscript.

In Section 3.2 and 4, natural variability of cloud microphysical properties is not sufficiently considered with liquid cloud and ice effective radius as well as COT and AOT assumed to be constant. This source of uncertainty should be examined, given VIS and SWIR channels are strong functions of these key properties, serving as main constraints for retrievals. Also, there is a lack of explanation on why these values are chosen. With a few constants assumed, the findings would only apply to a limited subset in nature.

It was not our intent to duplicate the work of Wang et al., 2022. Hence, we chose only a handful of cases to understand the variation caused by the smile; therefore we kept the cloud and aerosol micro-physical properties constant.

Why do you assume the surfaces to be barren or forest in Section 4 instead of snow and grass surfaces, which seem to cause large biases based on Fig. 5. Or why barren and forest are not taken into account in Fig.5, if they are important?

Due to the technical implementation, it is not possible to directly ingest surface albedo of grass in ECSIM. Instead, the high spectrally resolved BRDF is built following Vidot et al., 2014 there (Donovan et al. 2023). We see your point and therefore replaced birch and wheat for barren and evergreen broad-leaf forest in figure 5 to give the reader more context.

This part is not accounted for in sec. 4. MSI does not have enough bands to determine individual surface types. M-AOT and M-COP products rely on an auxiliary database mapping individual surface types. For details, see Docter et al. 2023 and Hünerbein et al. 2023, respectively.

While SMILE is not a new issue, it is an instrument specific issue. Hence, any investigations are specific to the band setting available for each individual instrument. Considering this instrument perspective, the only previous work available is of Wang et al., 2022. Our study intends to extend their study by investigating land surfaces instead of ocean surfaces and lower optical thickness than used in their study.

**Specific comments:**

Line 41: Please explain why you focus on smaller optical thickness. How small?
A higher impact and larger errors are expected due to the increasing spectral influence of land surface types. We include in brackets (OT<1).

Line 110: What is the value of "absolute accuracy requirement for aerosol optical thickness"? Please also add references for it.
Added reference to the mission requirement document (Wehr et al., 2006).

Line 131-135: The underlying physics that allows for retrievals and possible sources of uncertainties should be more clearly discussed here.
The underlying physics are described in Hünerbein et al. 2023. The reference has been added.

Line 138: What are the relevant assumptions in producing LUTs? You should provide a context for readers who may have little knowledge of this Level 2A retrieval. Please also provide references.
We modified: "MSI Level 2A aerosol and cloud retrievals" for clarification. The corresponding references have already been mentioned in the introduction: Hünerbein et al., 2023, 2024 and Docter et al., 2023. These publications of the EarthCARE special issue describe the relevant retrieval algorithms and assumptions for producing LUTs.

Line 140: Please summarize what is actually used from Baum et al. (2014).
Modified for clarification: "general habit mixture model". For more details, please see Hünerbein et al., 2024.

Line 141: You assume effective radius to be 10 and 5 microns for ice and liquid clouds, respectively. What is their natural variability shown in observations? How does that variability affect your results?
We clarified by adding: "These effective radii examples are not intended to represent the natural variability of all cloud observations."

Our focus in this study is to understand the SMILE for optically thin scatterer as mentioned in the introduction. Our understanding is that MSI should be able to observe background aerosol loadings, thin cirrus as well as fog (thin liquid clouds). Hence, it is legitimate to investigate limited examples in order to understand the effect the SMILE will have on these kind of measurements if it is ignored in Level 2A retrievals or it is not corrected for in Level 1. Please, see also our answer to Line 205.

[Figure]

Figure 9: To what extent the results would differ from what you have, if you assumed COT to be 50, or liquid cloud effective radius being 10 or 20 microns?

**Figure 1. The relative error of single scattering albedo due to the non-corrected smile effect for different effective radii for ice (reddish colors) and water clouds (bluish colors). The example is given for the SWIR-1 band.**

If we assume cloud with COT=50, the VIS band errors will decrease as the surface will play no important role anymore. To understand the influence due to the cloud effective radius. We added the figure above here. The effective radius values are set to 5 µm, 10 µm and 30 µm with an optical thickness of 10. The surface is not varied. The water and ice absorption channels (SWIR-1) are primarily a function of cloud particle size; this is reflected in the results. The relative error for the single scattering albedo getting stronger for larger cloud particle (**Error! Reference source not found.**).

While an increase in COT with constant effective radius will result in a decreasing relative error, an increase in effective radius assuming a constant COT will result in an increase in relative TOA normalized radiance error in only the SWIR1 band.

Added.

Our target was to understand the possible error sources caused by the SMILE. One gets a good first approximation using a COT of 10. The value has been chosen for convenience

and is representative according to ISCCP (online:
https://isccp.giss.nasa.gov/cloudtypes.html (last access: 09 February 2024)).

[Figure]

**Figure 2 International Satellite Cloud Climatology Project (ISCCP) cloud type definition in dependence on cloud optical thickness and cloud top pressure. Figure taken from: https://isccp.giss.nasa.gov/cloudtypes.html  (last access: 09 February 2024)**

Line 236-237: I can not see "cloud properties play a main role" in your results as you stated. Cloud effective radius was assumed to be constant in your experiment design. Please elaborate.
You are right. We removed this sentence.

Fig.12: Why are clouds not investigated over ocean surfaces?
This has already been done by Wang et al., 2022.

Line  252-253: what is the value of "a higher cloud effective radius"?
We clarified by adding (20 µm): "higher cloud effective radius (20 µm)"

**Technical corrections:**

Figures: colors of lines should be selected so that one line can be easily distinguished  from another. For instance, the colors you picked in figures 6, 7, and 9 to indicate dust, fmless and fmstrg are quite similar, which makes it difficult to read your graphs. Please change the colors.
Figures 6, 7, 9 along with 10, 11, 12 and 13 have been updated regarding their color schemes. We checked the figures with colorblindness simulator tools. Hence, colors were chosen in such a way that we avoid using reddish and greenish colors at the same time. Now, figures 12 and 13 show solid and dashed lines in order to make it easier for readers to distinguish them. Additionally, we associate bluish colors for clouds and darker yellow to

reddish colors for aerosols throughout all figures to make it easier for the reader to recognize different scatter types by their colors.

Fig. 1.: no color bar
Added.

Line 72: "Tab." is not a common abbreviation for table. Please use Table instead.
Done.

Line 78: "in the respective" => on
Done.

Figure 4. Please change the relative error on the y axis to values in %, so it will be consistent with your text. Also, it will be easier to compare the magnitude differences among the 4 channels. Same applies to other similar figures.
Updated for all figures that are showing the relative error of a quantity.

Line 105 to 106: The meaning of this sentence is unclear. Please rephrase.
Unnecessary dependent clause removed for clarity.

Line 121: "lower 10" => lower than 10
Done.

Line 127: This cannot be described as "same behavior", as you pointed out in the following sentences that magnitudes and even signs are different.
Rephrased to "a similar behavior".

Line 129: I think it should be Fig. 5d here.
Done.

Line 131-132: "VIS-to-NIR" => VIS to NIR
Removed.

Line 147: band => bands
Done.

Figure 6: What do "fmless" and "fmstrg" indicate, respectively?
Done.

Line 153: "While," => "While"
Done.

Line 164: 2 verbs
Updated to the originally intended two sentences.

Line 177: "it has not been accounted for gas absorption" => gas absorption is not accounted for (if I understand correctly).
Correct. Modified.

Line 180: "Exemplary" => For example
Done.

Line 206: "are altering" => vary
Done.

Line 217: "of e.g." => such as
Done.

Line 231: "optical" => optically
Done.

Line 239: "implication" => implications
Done.

"aerosol" => aerosols
Not modified because it refers to aerosol optical thickness, not "aerosols optical thickness".

Line 240: Please rephrase the 1st sentence of this paragraph.
Rephrased

Line 249: what is "TIR"? Not previously mentioned.
Modified to: thermal infrared (TIR)

Line 287: "mitigate already existing retrieval algorithms" => mitigate the relevant errors in existing retrieval algorithms.
Errors are not directly mitigated. Rather retrieval assumptions are mitigated to include the smile. We modified the sentence to: "to consider the smile effect: ..." to be more precise.

---

## Author Comment (AC2)

**Reply to Anonymous Referee #2**

We thank the reviewer for their helpful and constructive comments. Please find our answers (black text) to the comments (blue text) in-line below. Respective changes are indicated in the revised manuscript in blue and are stated here in addition when a reviewer's comment leads to a substantial modification of the manuscript along with the updated line number, where necessary.

**General comments:**

This manuscript discusses the impact of the Spectral Misalignment Effect (SMILE) on the EarthCARE Multi-spectral Imager (MSI) in retrieving aerosol and cloud properties. The paper is well suited to the scope of AMT. I suggest some minor revision considering the potential improvements as follows.

Generally, the colors in some figures are not clear enough to let readers to distinguish one from another. For example, the light-yellow line in Figure 5 is very hard to see, and similarly, the light-yellow lines that represents salt in Figure 6, 7 and 9 are seems easily mixed with other color lines. Besides, lines in Figure 12 and Figure 13 (b) are all in reddish colors, which color arrangement should be considered. Finally, the dark purple color in the bottom area of Figure 10 (c) makes the words "water" almost invisible, you should consider change the color into lighter bluish color, or change the color of the words "water" into white.

Figures 5, 6, 7, 9, 10, 11, 12 and 13 have been updated regarding their color schemes. We checked the figures with colorblindness simulator tools. Hence, colors were chosen in such a way that we avoid using reddish and greenish colors at the same time. Now, figures 12 and 13 show solid and dashed lines in order to make it easier for readers to distinguish them. Additionally, we associate bluish colors for clouds and darker yellow to reddish colors for aerosols throughout all figures to make it easier for the reader to recognize different scatter types by their colors.

**Specific comments:**

Line 31-35: Please add some reference to support the story of "mitigation strategies have been implemented by ESA and industry".
Reference added. For details, see page 27 therein.

Figure 1: There is no color bar to explain the range of the MSI response functions shown, please add it.
Done.

Line 141-142: You noted the effective radii of both types of cloud droplets, how about the definition of optical thickness? Are they as same as you noted in Line 181-182?
Yes, that is correct. However, since it is only needed for figure 9, we only mention it when describing this figure. Please note that the normalized extinction used here is a relative quantity describing the spectral behavior of the scatterer. We added "normalized" to be more specific.

Line 178: "level2" --> "Level 2", "retrieval" --> "retrievals"
Done.

Line 185-186: Can you provide some references for this?
We added a reference to Seidel and Popp, 2012.

Line 205: I think you should at least add another case with a larger COT for the water cloud, only 10 is not sufficient to represent all water clouds.
We agree that COT=10 is not representative of all water clouds. It was not our intent to represent all clouds. We would and do refer to Wang et al. for higher OTs than 10 since they did a thorough analysis.